# Multi-Objective Guidance via Importance Sampling for Target-Aware Diffusion-based *De Novo* Ligand Generation

**Julian Cremer** [* 1 2]  **Tuan Le** [* 1 3]  **Frank Noé** [4]  **Djork-Arné Clevert** [1]  **Kristof Schütt** [1]

## Abstract

The generation of ligands that both are tailored to a given protein pocket and exhibit a range of desired chemical properties is a major challenge in structure-based drug design. Here, we propose an in-silico approach for the *de novo* generation of 3D ligand structures using the equivariant diffusion model PILOT, combining pocket conditioning with a large-scale pre-training and property guidance. Its multi-objective trajectory-based importance sampling strategy is designed to direct the model towards molecules that not only exhibit desired characteristics such as increased binding affinity for a given protein pocket but also maintains high synthetic accessibility. This ensures the practicality of sampled molecules, thus maximizing their potential for the drug discovery pipeline. PILOT significantly outperforms existing methods across various metrics on the common benchmark dataset CrossDocked2020. Moreover, we employ PILOT to generate novel ligands for unseen protein pockets from the Kinodata-3D dataset, which encompasses a substantial portion of the human kinome. The generated structures exhibit predicted $IC_{50}$ values indicative of potent biological activity, which highlights the potential of PILOT as a powerful tool for structure-based drug design.

## 1. Introduction

One of the innovative machine learning techniques increasingly employed in structure-based drug discovery (SBDD) is the application of generative diffusion models. Originally utilized in fields like computer vision and natural language processing, these models also excel in capturing the complex patterns of 3D molecular structures, particularly when enhanced with features that reflect the symmetry and specific target-related characteristics of proteins (Green et al., 2021; Luo et al., 2021; Ragoza et al., 2022; Liu et al., 2022; Tan et al., 2022; Peng et al., 2022; Powers et al., 2023; Luo et al., 2021; Peng et al., 2022; Guan et al., 2023; Schneuing et al., 2023). The effectiveness of these models hinges on training with detailed protein structures, which allows for the generation of ligands that are not only structurally compatible but also specifically designed for interaction with target proteins. However, 3D generative models often yield ligands with sub-optimal drug-like qualities, characterized by a high prevalence of fused rings and low synthetic accessibility (Xia et al., 2024; Schneuing et al., 2023; Guan et al., 2023). While generated ligands fit well in a protein binding pocket, these methods lack a mechanism to guide the generative process towards ligands with desired chemical properties such as binding affinity, stability, or bioavailability (Gómez-Bombarelli et al., 2018).

In this study, we introduce PILOT (**P**ocket-**I**nformed **L**igand **Op**timization) – an equivariant diffusion model designed for *de novo* ligand generation with property guidance. We employ importance sampling to replace less desirable intermediate samples with more favorable ones, thus re-weighting trajectories during the generative process. This strategy enables the use of any pre-trained, unconditioned diffusion score model for sampling, which is then enhanced by integrating the capabilities of a surrogate model, similar to classifier guidance (Dhariwal & Nichol, 2021). However, while classifier guidance may drive the sampling trajectory to adversarial, out-of-distribution structures (Dhariwal & Nichol, 2021), trajectory re-weighting ensures that samples remain within distribution. Additionally, backpropagation is not required. As a result, importance sampling is significantly faster, especially for ligand-pocket complexes. As trajectory re-weighting can be conducted in parallel for multiple properties, we focus on three critical properties for drug discovery: synthetic accessibility (SA), docking scores and potency ($IC_{50}$). Our findings demonstrate that PILOT generates ligands that not only exhibit a significant improvement in synthesizability and drug-likeness but also achieve favorable docking scores.

---

[*]Equal contribution  [1]Pfizer Research & Development [2]University Pompeu Fabra [3]Freie Universität Berlin [4]Microsoft Research. Correspondence to: Tuan Le <tuan.le@pfizer.com>, Julian Cremer <julian.cremer@pfizer.com>.

*Accepted at the 1st Machine Learning for Life and Material Sciences Workshop at ICML 2024.* Copyright 2024 by the author(s).

## 2. Methods

### 2.1. PILOT

In this study, we aim to generate novel molecules $M$ *de novo*, conditioned on a protein pocket $P$ while addressing multiple objectives $c$, such as synthetic accessibility, docking score, and predicted half-maximal inhibitory concentration ($IC_{50}$). We adopt the EQGAT-diff model (Le et al., 2024) to implement $p_\theta(M|P)$. For all implementation and model details we refer to Le et al. (2024).

### 2.2. Importance Sampling for Property Guidance

To sample ligands from the distribution $p_\theta(M|P,c)$, we utilize Bayes' theorem to decompose the probability density into $p_\theta(M|P,c) \propto p_\delta(c|M,P)p_\theta(M|P)$. We further assume that the multiple properties $c = (c_1, c_2, \ldots, c_k)$ are conditionally independent, leading to the factorization $p_\delta(c|M,P) = \prod_{l=1}^{k} p_{\delta_l}(c_l|M,P)$. To accurately predict these conditions, we train $p_\delta(c|M,P)$ as $p_\delta(c|M_t,P,t)$ along the forward noising diffusion trajectory, where $M_t$ represents the state of the ligand at time $t$. The rationale behind this training approach is that denoising steps closer to the original data distribution retain a clearer signal of the input ligand, making them highly informative. In contrast, steps closer to the prior noise distribution, although less informative, can still provide valuable discriminative insights for $p_\delta$. This strategy leverages the nuanced progression of information degradation during the diffusion process to efficiently guide the generation of desired ligands without mode collapse. The property model $p_\delta$ is trained using the mean squared error and cross-entropy loss for continuous and discrete properties, respectively. The importance weights $w_k$ for each intermediate sample $M_{t,k}$ are determined by performing a softmax normalization across the set of (noisy) ligands, as outlined in Algorithm 1.

## 3. Results and discussion

### 3.1. Multi-objective de novo generation using importance sampling

In previous studies, utilizing 3D target-aware molecule generation led to a significant challenge: the poor synthetic accessibility (SA) of the generated molecules. These models often produce molecules with complex, fused, and uncommon ring systems, which are difficult to synthesize (Xia et al., 2024; Guan et al., 2023; Schneuing et al., 2023). This issue underscores the need for approaches that not only produce molecules with strong binding affinities but also ensure that these molecules can be feasibly synthesized. To address this, we propose a trajectory-based importance sampling method that utilizes property-specific expert models. These surrogate models must be able to predict the properties of

---

**Algorithm 1** Importance sampling for property-guided ligand generation

**Input:** Pocket $P$, condition $c$, number of ligands $K$, $\tau$ temperature, every importance step $N$, diffusion model $p_\theta$ and property models $p_\delta$.

**Output:** Generated ligands $\{M_i\}_{i=1}^{K}$ conditioned on $(P,c)$.

1: Sample $K$ ligands from prior distribution $M_T \sim N(0,I) \times C(\hat{p}_c)$
2: **for** $t = T-1, \ldots, 1$ **do**   ▷ Run reverse diffusion trajectory
3:   Sample $M_{t-1} \sim p_\theta(M_{t-1}|M_t,P)$
4:   **if** $t \mod N = 0$ **then**   ▷ Importance step
5:    **for** $k = 1, \ldots, K$ **do**
6:     $c_k = p_\delta(c_k|M_{k,t-1},P)$ ▷ Property prediction
7:    **end for**
8:    Importance weight computation based on population $\{(M_{k,t-1},c_k)\}_{k=1}^{K}$, here max $c$:
9:    $w_k = \frac{\exp(c_k/\tau)}{\sum_{j=1}^{K} \exp(c_j/\tau)}$
10:    Draw new population with replacement:
11:    $\{M_{k,t-1}\}_{k=1}^{K} \sim \text{Multinomial}(\{M_{k,t-1}\}, \{w_k\})$
12:   **end if**
13: **end for**
14: return $\{M_{k,0}\}_{k=1}^{K}$

---

interest at any step of the diffusion trajectory, similar to classifier-guidance (Dhariwal & Nichol, 2021).

As properties such as synthetic accessibility are determined solely based on the ligand, whereas others, like docking scores, depend on the interaction between the ligand and the protein pocket, suitable property predictors $p_{\delta_i}$ may be defined as required. During the sampling process of a set of $K$ noisy ligands $\{M_1, M_2, \ldots, M_K\}$, we apply importance weights derived from $p_\delta(c|M,P)$ to rank each noisy sample at its current position in the state space. This process is inspired by the Sequential Monte Carlo (SMC) method (Doucet et al., 2001; Del Moral et al., 2006). Since the reverse diffusion trajectory is inherently stochastic, our goal is to preferentially select those samples most likely to follow a trajectory that results in ligands meeting the specified conditions $c$, as schematically depicted in Fig. 1. To accurately predict these conditions, we train $p_\delta(c|M,P)$ as $p_\delta(c|M_t,P,t)$ along the forward noising diffusion trajectory, where $M_t$ represents the state of the ligand at step $t$. A similar replacement strategy has previously been applied by Trippe et al. (2023) and Wu et al. (2023) in the context of diffusion models for protein backbone modeling and motif scaffolding.

The evaluation of the importance sampling approach is performed for both single- and multi-objective optimization scenarios, focusing on SA and docking score guidance. We refer to guidance with an SA score model as *SA-conditional*

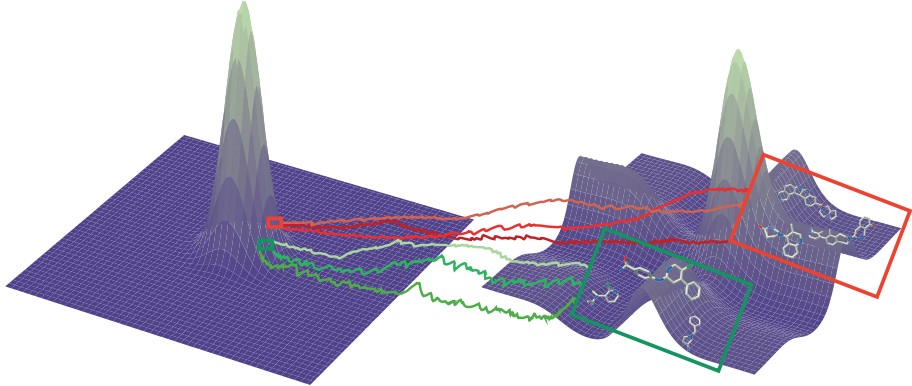

*Figure 1.* Visualization of the importance sampling algorithm. The shape of the prior (left) and target (right) distribution, where ligands at the target distribution are highlighted in two different regions based on a property function, which is synthetical accessibility in this case. At $t = T$ (left), noisy samples are drawn from the prior, and during the reverse trajectory, stochastic paths that lead to promising candidates are selected and de-noised in state-space to converge to samples from the data distribution at $t = 0$ (right). Ligands in the green box refer to molecules with high synthetic accessibility according to SA score, while molecules in the red box refer to rather inaccessible ones.

and using a docking-score model as *docking-conditional*. When both objectives are considered, we refer to the model as *SA-docking-conditional*. In each case, the unconditional base model is augmented with the respective property model during the sampling process. Table 2 shows the correlation matrix of the CrossDocked2020 dataset. The SA scores exhibit a negative correlation with ligand size, i.e. larger molecules tend to be less synthetically accessible on average. Conversely, the positive correlation between SA scores and QED suggests that molecules with higher QED are generally more synthetically accessible. Docking scores show a strong negative correlation with both the number of rings and the number of atoms. This implies that models driven by docking scores tend to generate larger molecules with more (fused) rings. However, such molecular characteristics typically result in decreased SA scores and QED, presenting a trade-off between optimizing for docking affinity and maintaining synthetic feasibility. By incorporating these insights into our modeling approach, we aim to balance the dual objectives of binding efficacy and synthetic accessibility, thereby enhancing the practical utility of the generated molecules in drug discovery.

Table 1 shows that our model reproduces the observed correlations of the dataset. When guiding the unconditional model with the SA score, we notice a significant enhancement not only in the SA score, which increases to 0.77, but also improvements in QED and Lipinski's rule of five compliance. The mean docking scores remain consistent with those of the unconditional model. However, there is a notable reduction of docking performance in the top-10 ligands, consistent with the correlations observed in the CrossDocked dataset. Conversely, applying docking-score guidance exclusively results in diminished SA scores and QED, while the docking scores themselves markedly in-

crease. This reflects the trade-offs involved in optimizing for docking efficacy at the expense of synthetic accessibility and drug-likeness. When applying both SA and docking-score guidance, the model achieves comparably high values for SA, QED, and Lipinski, while significantly improving docking scores and outperforming TargetDiff by a large margin across metrics.

To mitigate the adverse impact on SA scores and drug-likeness typically associated with high docking scores of larger molecules, we introduce a normalization strategy where docking scores are adjusted by the square root of the number of atoms per ligand. The results of this adjusted model, denoted as SA-docking-conditional (norm), are presented in the final row of Table 1. Here, we observe a significant increase in docking scores compared to the unconditional model, while the SA scores improve to 0.78, compared to 0.77 in the SA-conditional model. This illustrates how our multi-objective optimization strategy balances different property demands. Such balanced outcomes are critical for advancing the practical utility of generated molecules in drug discovery, ensuring that they not only bind effectively but are also feasible for synthesis.

Fig. 3 illustrates the evolution of the sample space across the unconditional, SA-conditional, docking-conditional, and SA-docking-conditional models. Each plot in this figure includes a red rectangle that identifies the regions where samples exceed the respective means of the test set, indicating improved property scores. The first row of Fig. 3 compares the drug-likeness (QED) of sampled ligands with their synthetic accessibility (SA) scores. The SA-conditional model shows a notable shift with most of the sample mass residing within the red rectangle. Thus, it successfully generates samples with notably higher SA scores compared

|  | # Rings | # Rotatable bonds | # Atoms | Docking scores | QED | SA |
|---|---|---|---|---|---|---|
| # Rotatable bonds | 0.14 | | | | | |
| # Atoms | 0.75 | 0.66 | | | | |
| Docking scores | -0.71 | -0.23 | -0.67 | | | |
| QED | 0.03 | -0.50 | -0.33 | -0.10 | | |
| SA | -0.28 | -0.27 | -0.43 | 0.16 | 0.42 | |
| logP | 0.45 | 0.08 | 0.35 | -0.55 | 0.36 | 0.29 |

*Figure 2.* Correlation matrix that includes the number of rings, number of atoms, docking scores, quantitative estimate of drug-likeness (QED), and synthetic accessibility (SA) scores using the CrossDocked training set.

*Table 1.* Performance comparison among unconditional sampling, SA-conditional, docking-conditional, and SA-docking-conditional sampling using the CrossDocked test set, which includes 100 targets. For each target, 100 ligands were sampled. We assessed the performance based on several criteria: mean docking scores obtained from QVina2 re-docking, the top-10 mean docking scores per target, drug-likeness (QED), synthetic accessibility score (SA), compliance with Lipinski's Rule of Five (Lipinski), and mean diversity (Diversity) across targets and ligands.

| Model | QVina2 (All) ↓ | QVina2 (Top-10%) ↓ | QED ↑ | SA ↑ | Lipinski ↑ | Diversity ↑ |
|---|---|---|---|---|---|---|
| Training set | $-7.57_{\pm2.09}$ | - | $0.53_{\pm0.20}$ | $0.75_{\pm0.10}$ | $4.57_{\pm0.91}$ | - |
| Test set | $-6.88_{\pm2.33}$ | - | $0.47_{\pm0.20}$ | $0.72_{\pm0.13}$ | $4.34_{\pm1.14}$ | - |
| TargetDiff | $-7.32_{\pm2.47}$ | $-9.67_{\pm2.55}$ | $0.48_{\pm0.20}$ | $0.58_{\pm0.13}$ | $4.59_{\pm0.83}$ | $\mathbf{0.75}_{\pm0.09}$ |
| unconditional | $-7.33_{\pm2.19}$ | $-9.28_{\pm2.26}$ | $0.49_{\pm0.22}$ | $0.64_{\pm0.13}$ | $4.40_{\pm1.05}$ | $0.69_{\pm0.07}$ |
| SA-conditional | $-7.32_{\pm2.25}$ | $-8.91_{\pm2.29}$ | $\mathbf{0.58}_{\pm0.19}$ | $\mathbf{0.77}_{\pm0.10}$ | $\mathbf{4.82}_{\pm0.54}$ | $0.73_{\pm0.08}$ |
| docking-conditional | $\mathbf{-9.17}_{\pm2.48}$ | $\mathbf{-10.94}_{\pm2.51}$ | $0.54_{\pm0.13}$ | $0.62_{\pm0.08}$ | $4.70_{\pm0.41}$ | $0.57_{\pm0.10}$ |
| SA-docking-conditional | $-8.35_{\pm2.75}$ | $-10.36_{\pm2.62}$ | $\mathbf{0.58}_{\pm0.17}$ | $0.72_{\pm0.12}$ | $\mathbf{4.88}_{\pm0.44}$ | $0.68_{\pm0.09}$ |
| SA-docking-conditional (norm) | $-7.92_{\pm2.44}$ | $-9.85_{\pm2.33}$ | $0.56_{\pm0.19}$ | $\mathbf{0.78}_{\pm0.11}$ | $\mathbf{4.84}_{\pm0.47}$ | $\mathbf{0.75}_{\pm0.13}$ |

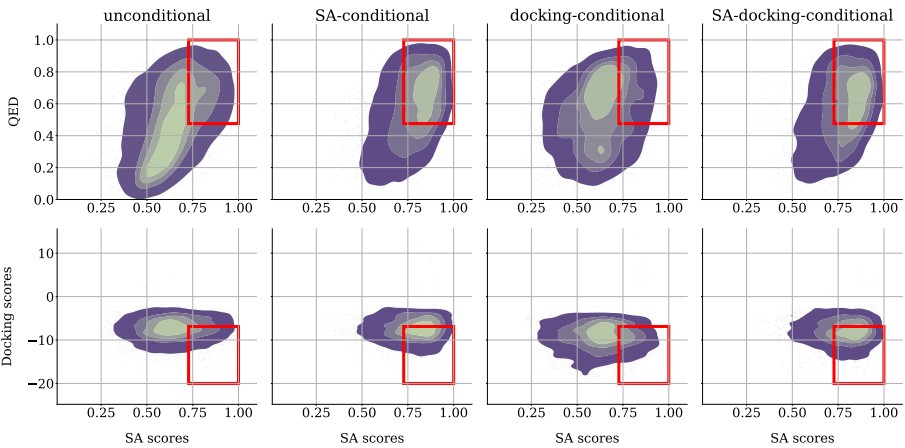

*Figure 3.* Scatter plots with Gaussian kernel density estimation (KDE) were used to illustrate the evolution of QED, SA, and docking scores for all sampled ligands across test targets for different sampling methods: unconditional, SA-conditional, docking-conditional, and SA-docking-conditional sampling. Red rectangles within these plots highlight regions where sampled ligands demonstrate superior QED, SA, and docking scores compared to the test set. **Top**: Relationship between QED and SA scores. **Bottom** Relationship between docking scores and SA scores.

to both the unconditional model and the test set ligands, while largely preserving docking scores. In contrast, the docking-conditional model exhibits lower docking scores on average at the expense of the SA scores. The SA-docking-conditional model demonstrates a good balance, transitioning towards both high SA scores and low docking scores. Remarkably, most of the sampled ligands from this model not only fall within the red rectangle but also significantly surpass the test set ligands in terms of docking scores with equal SA scores as listed in Table 1, while the model with normalization improves in both metrics.

Our findings demonstrate that using importance sampling as a guidance mechanism in the diffusion model is a potent

*Table 2.* Performance comparison among unconditional and $pIC_{50}$-conditional sampling using the Kinodata-3D test set, which includes 10 targets. For each target, 100 ligands were sampled. We assessed the performance based on several criteria: mean docking scores obtained from QVina2 re-docking, the top-10 mean docking scores per target, (predicted) $pIC_{50}$, drug-likeness (QED), synthetic accessibility score (SA), compliance with Lipinski's Rule of Five (Lipinski), and mean diversity (Diversity) across targets and ligands.

| Model | Vina (All) ↓ | Vina (Top-10%) ↓ | $pIC_{50}$↑ | QED ↑ | SA ↑ | Lipinski ↑ | Diversity ↑ |
|---|---|---|---|---|---|---|---|
| Training set | $-9.20_{\pm1.13}$ | - | $7.05_{\pm1.28}$ | $0.49_{\pm0.16}$ | $0.75_{\pm0.07}$ | $4.73_{\pm0.52}$ | - |
| Test set | $-8.78_{\pm1.13}$ | - | $6.41_{\pm1.56}$ | $0.61_{\pm0.14}$ | $0.79_{\pm0.05}$ | $\mathbf{4.96}_{\pm0.22}$ | - |
| unconditional | $-8.49_{\pm1.05}$ | $-9.79_{\pm0.87}$ | $6.28_{\pm0.68}$ | $\mathbf{0.63}_{\pm0.14}$ | $\mathbf{0.75}_{\pm0.13}$ | $4.95_{\pm0.25}$ | $\mathbf{0.65}_{\pm0.06}$ |
| $pIC_{50}$-conditional | $-8.60_{\pm0.98}$ | $-9.75_{\pm0.86}$ | $\mathbf{7.65}_{\pm0.78}$ | $0.62_{\pm0.16}$ | $0.67_{\pm0.09}$ | $4.94_{\pm0.28}$ | $0.57_{\pm0.06}$ |

strategy for steering the generation of molecules towards desired regions of chemical space. The method effectively modifies molecular properties to align with desired multi-objective property profiles without the need for any additional backpropagation, albeit within the constraints of the data distribution used for training.

### 3.2. Kinodata-3D

Kinodata-3D (Backenköhler et al., 2024) is an *in silico* curated and processed collection of kinase complex cross-docked data designed to facilitate the training of machine learning models on structural protein-ligand complexes with experimental binding affinity data. Despite the significant advances in virtual screening and the widespread use of docking as a tool for evaluating ligand efficacy, the correlation between docking scores and experimental binding affinities, e.g. measured by the half maximal inhibitory concentration $IC_{50}$, remains weak at best. Consequently, reliance on docking scores as stand-ins for binding affinities is potentially misleading. To address this challenge, we apply our guidance mechanism to directly utilize experimental binding affinities. We leverage the Kinodata-3D dataset, annotated with experimental $pIC_{50}$ values, to train PILOT on ligand-kinase complexes. Simultaneously, we train a property model predicting $pIC_{50}$, to guide the diffusion model with the propose importance sampling towards ligands that are more likely to be potent inhibitors. This strategy aims to reduce the reliance on less accurate proxies such as docking scores. We evaluate the models on a hold-out test set comprising ten kinase targets that were not included in either the training or validation datasets. The performance of our $pIC_{50}$-conditional model is summarized in Table 2. The $pIC_{50}$-conditional model shows a significant improvement in predicted mean $pIC_{50}$ values of $7.65_{\pm0.78}$ compared to the test set ligands ($6.41_{\pm1.56}$). At the same time, it maintains robust performance metrics in terms of docking scores and other critical properties such as QED, SA-score, and compliance with Lipinski's rule of five. Note, that the current approach is limited since $pIC_{50}$ values are inherently noisy, in particular when collected across various data sources.(Landrum & Riniker, 2024) Thus, the predicted

binding affinities should be interpreted cautiously.

## 4. Conclusions

We have introduced PILOT, a novel equivariant diffusion-based model tailored for *de novo* ligand generation conditioned on protein pockets in three-dimensional space. Our research demonstrates the superior performance of PILOT compared to existing state-of-the-art models in this domain, as evidenced by a comprehensive evaluation across a spectrum of metrics critical in medicinal chemistry and drug design. We have proposed a trajectory-based importance sampling strategy, which enables targeted steering of ligand generation towards desired chemical properties. This technique guides the generation process towards ligands with properties such as synthetic accessibility, drug-likeness, docking scores and potency by using surrogate models. This strategy represents a significant advancement in structure-based drug discovery, offering researchers a powerful tool to design molecules with tailored properties. The dependency on the availability and quality of training data remain a critical challenge for deploying AI models like PILOT in drug discovery pipelines. In the domain of structure-based drug design, data can often be sparse, noisy, and of varying quality, which significantly impacts the learning and predictive capabilities of ML models. While our method relies heavily on surrogate models and proxies like the RDKit synthetic accessibility (SA) scores to estimate the synthesizability of generated ligands, these scores may not fully capture the complexities and practical challenges of synthetic chemistry. Addressing these challenges will require a concerted effort to enhance data collection practices, improve data quality, and expand the variety of data sources. Moving forward, we see potential applications of PILOT in the drug discovery pipeline by integrating this model with other AI-driven tools and technologies, such as automated synthesis platforms and high-throughput screening to accelerate drug design. Furthermore, the scope of our model may be extended from small molecule drugs to biologic therapeutics involving for example peptides or antibodies.

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
