# OpenReview forum: "Multi-Objective Guidance via Importance Sampling for Target-Aware Diffusion-based De Novo Ligand Generation"
_ICML.cc/2024/Workshop/ML4LMS — ML4LMS Poster_

### Official Review · Reviewer_T71M · 2024-06-08
**Interesting application of published concepts, but some justifications and experimental details missing**

**Rating:** 6
**Confidence:** 4

**Review:**

1.⁠ ⁠Overall evaluation

Opinion: The authors adapt a previously used SMC scheme to a novel application and guide the model in clever ways to improved properties. While some of their claims do not seem supported by the data (see below) and some details about their method are missing, I think the idea itself is interesting to the community, which is why I vote for a weak accept.

Summary: The authors propose an SMC-inspired importance sampling scheme for protein-conditioned ligand generation. They evaluate the performance of their method on the widely used CrossDocked dataset as well as on the Kinodata-3D dataset and show that their model achieves a better trade-off between the molecular properties under consideration.

Contributions:

[C1]  Apply an SMC algorithm previously used in protein design to ligand design and adopt it to this setting.
[C2]  Benchmark the performance of their algorithm on different datasets and via a discussion of different metrics.

2.⁠ ⁠Strengths & Weaknesses:

[S1] the discussion about metrics is nuanced, highlighting the important trade-offs between the metrics under consideration
[S2] the application of this strategy to ligand design is novel (although it has been used fro protein design in the past) and the authors demonstrate its benefit, making it a useful addition to the toolkit of the community.

[W1] The number of particles/samples that are evaluated and reweighted during one diffusion step is a crucial parameter of SMC methods that allows the practicioner to trade off between quality and computational budget. I did not find numbers on how many samples were chosen here nor any computational resource numbers. Ablations on the impact of different numbers of particles would be useful to properly judge the performance of the method here.
[W2] The performance of their model on Kinodata-3D is measured by an oracle model that was also trained by the authors. Before using this as a metric, it should be validated that this model is actually any good at predicting pIC50 values. While the authors acknolwedge that limitation, this is the only strength their method shows on this dataset, therefore an independent validation of their oracle would be helpful to strengthen their claim here.

---

### Official Review · Reviewer_qLvY · 2024-06-12
**good paper maybe**

**Rating:** 7
**Confidence:** 3

**Review:**

I wonder if it would be possible to also include comparisons like summary of different properties of reference molecules and molecules generated by PILOT vs other methods like TargetDiff or Pocket2Mol. Also maybe including binding affinity prediction results on PDBbind v2020 using your method + EGNN ( for example) like in the targetDiff paper to demonstrate the superiority of the proposed method.

---

### Official Review · Reviewer_QVgS · 2024-06-12
**Gradient-free guidance for property enhancement with pocket-conditioned molecule diffusion models**

**Rating:** 8
**Confidence:** 4

**Review:**

This paper proposes and evaluates a guidance strategy for pocket-conditioned molecule diffusion models. The guidance strategy is based on importance sampling similar to Sequential Monte Carlo guidance strategies previously proposed for protein diffusion models. It adapts these strategies in a way that removes the need for classifier gradients, which is useful when dealing with discrete variables. The evaluation of the effects of property guidance is valuable and demonstrates that guided sampling can lead to improvements in property values (although further clarity on the evaluation setup would better help contextualise the reported results, see below).

Overall this paper addresses a setting that is of clear interest to the community, and proposes a sensible strategy for addressing an important problem with apparently quite promising results. However further clarity on the evaluation setup is required to properly assess the significance of the results.
* the discussion of the weaknesses of classifier guidance is unconvincing. Classifier guidance should be considered as a baseline; if this is not possible due e.g. to discreteness, this should be clearly explained.
* how does the trajectory-based importance sampling guidance compare to naive importance sampling, where samples from the unconditional model are reweighted based on property values computed at the end of the trajectory?
* I wasn't able to figure out whether the 100 samples for the guided sampling strategies came from running a single importance sampling trajectory with K=100, or from running 100 trajectories with some other K. In the latter case, unconditional sampling should be allowed Kx100 samples to match the compute budget, and it'd be interesting to know the diversity within the trajectory. In the former case, everything look good!
* what is the training set used to train the property predictors for the results in Table 1?

The authors also occasionally make statements that could lead to confusion about the contribution: PILOT is NOT a novel diffusion model. It is a guidance strategy. The former impression is (potentially misleadingly) given by sentences like:

'We introduce PILOT - an equivariant diffusion model designed for de novo ligand generation with property guidance.'
'We have introduced PILOT, a novel equivariant diffusion-based model...'